# Cas Endonuclease Technology—A Quantum Leap in the Advancement of Barley and Wheat Genetic Engineering

**DOI:** 10.3390/ijms20112647

**Published:** 2019-05-29

**Authors:** Iris Koeppel, Christian Hertig, Robert Hoffie, Jochen Kumlehn

**Affiliations:** Plant Reproductive Biology, Leibniz Institute of Plant Genetics and Crop Plant Research (IPK) Gatersleben, 06466 Seeland, Germany; koeppel@ipk-gatersleben.de (I.K.); hertig@ipk-gatersleben.de (C.H.); hoffier@ipk-gatersleben.de (R.H.)

**Keywords:** cereals, CRISPR, crops, genetic engineering, genome editing, plant, *Triticeae*

## Abstract

Domestication and breeding have created productive crops that are adapted to the climatic conditions of their growing regions. Initially, this process solely relied on the frequent occurrence of spontaneous mutations and the recombination of resultant gene variants. Later, treatments with ionizing radiation or mutagenic chemicals facilitated dramatically increased mutation rates, which remarkably extended the genetic diversity of crop plants. However, a major drawback of conventionally induced mutagenesis is that genetic alterations occur simultaneously across the whole genome and at very high numbers per individual plant. By contrast, the newly emerging Cas endonuclease technology allows for the induction of mutations at user-defined positions in the plant genome. In fundamental and breeding-oriented research, this opens up unprecedented opportunities for the elucidation of gene functions and the targeted improvement of plant performance. This review covers historical aspects of the development of customizable endonucleases, information on the mechanisms of targeted genome modification, as well as hitherto reported applications of Cas endonuclease technology in barley and wheat that are the agronomically most important members of the temperate cereals. Finally, current trends in the further development of this technology and some ensuing future opportunities for research and biotechnological application are presented.

## 1. Introduction

### 1.1. Historical View of Genetic Modification in Crop Plants

For more than 10,000 years, plants have been cultivated to feed humans and their livestock, as a source of raw materials and to generate energy. Over time, the driving forces of evolution, domestication, and breeding of cultivated plants have changed according to environmental conditions and societal demands. To cope with the growth of the world’s population and the consequent increase in demand for food, the total production needs to be substantially increased during the next decades [1,2]. Not only is the growing world population a challenge, but also the expected climatic changes [3,4]. Thus, the plants of the future also have to be better adapted to harsh weather conditions [5,6]. In addition, the demands of modern society are increasing; food should be of better and consistent quality, healthier, and more diverse. At the same time, society calls for a substantially reduced application of fertilizers and pesticides to render agriculture more sustainable [2].

In nature, mutations occur spontaneously and alterations that provide an advantage under the given conditions are likely to prevail over time. For modern plant breeding, however, the rate of spontaneous mutations is too low to keep up with the demands for crop improvements. Moreover, it is in the nature of spontaneous mutations that they are unpredictable in terms of both position and resultant nucleobase sequence. In the 1930s, mutation breeding emerged, in which mutations are deliberately induced with chemicals such as ethylmethanesulfonate (EMS) or with ionizing radiation. Induced mutagenesis breeding has so far produced over 3000 approved varieties [7] and for many further cultivars it has not been documented which induced mutations they have inherited from germplasms they derive from. Using this technology, however, thousands of mutations occur at different sites in a single plant’s genome at a time. Therefore, a vast number of undesirable mutations have to be eliminated via cumbersome and time-consuming back-crossing procedures [5]. The method of Targeted Induced Local Lesions in Genomes (TILLING) represented a significant progress in the detection of mutations. With TILLING or Eco-TILLING, respectively, it is possible to readily identify induced and spontaneous mutations in known genes of interest [8,9].

The method of gene transfer using *Agrobacterium tumefaciens* was introduced in the 1980s [10]. Ever since, genes derived from unrelated organisms or genetically modified gene variants can be introduced into plant genomes. This technique makes it possible to modify or introduce performance-determining genes into cultivated plants more quickly and in a more targeted manner. However, there are some limitations, such as the fact that the transferred gene is integrated at a random position in the genome and that changes to the plant’s own genes are only possible to a very limited extent. The insertion site in the genome then also determines whether and how effectively the introduced gene is expressed, and thus to what extent the respective trait will be modified. In addition, not all plants can be transformed equally well by the use of *Agrobacterium*, with cereal species being a particular challenge in this respect [11].

With the new methods for targeted genome modification that are based upon customizable endonucleases, it is possible to modify DNA sequences at a previously defined site of choice in the host genome. The standard application of this technology is site-directed mutagenesis, in which the location of the mutation can be precisely determined, whereas the resulting nucleobase sequence is random [12]. However, the precision can be increased by a variety of approaches. The most sophisticated one involves the use of an artificial DNA repair template implemented via homology-directed DNA repair. By this still very challenging principle, any sequence of choice can be created at any predefined genomic locus. 

Wheat (*Triticum aestivum*) and barley (*Hordeum vulgare*) are among the most important cereals in the world. Among all crops, wheat occupies the largest cultivated area in the world, and barley is one of the oldest domesticated crops, currently being the fourth most frequently cultivated cereal (http://faostat/fao.org). With a size of about 17 Gbp, wheat has a very large and complex genome. Only recently has the wheat genome been almost completely read out and this data has been made publicly available [13]. Hexaploid bread wheat evolved via hybridizations from several ancestors. This process involved allopolyploidization, resulting in a total of three diploid subgenomes [14]. In contrast to wheat, barley has a genome with a size of 5.1 Gbp. Already in 2012, a detailed draft of the barley genome was published [15], which was recently complemented by much improved data, including genomic sequences of a large number of representative gene bank accessions [16,17].

### 1.2. Platforms of Customizable Endonucleases

In plant research and biotechnology, four platforms of customizable endonucleases have been used so far; meganucleases, zinc-finger nucleases (ZFNs), transcription activator-like effector nucleases (TALENs), and the RNA-guided, clustered, regularly interspaced, short palindromic repeats (CRISPR)-associated (Cas) endonucleases. 

Meganucleases are naturally occurring endonucleases that recognize and cut comparatively long (>12 bp) DNA sequences. The most commonly used meganuclease is the I-*Sce*I from *Saccharomyces cerevisiae* [18,19]. The ability of this endonuclease to induce double-strand breaks in plants was first demonstrated by Puchta et al. [20] using *Nicotiana* protoplasts. Meganucleases are very specific and efficient. It is, however, very difficult to reengineer these enzymes for target sequences other than their native ones [21]. Hence, their use is fairly restricted as compared to the alternative endonuclease platforms [22,23]. As a consequence, meganucleases have been used in plants almost exclusively for basic research, in particular for DNA repair mechanisms [24]. 

Zinc-finger nucleases are hybrid proteins with a DNA recognition domain consisting of at least three zinc-fingers combined with a *Fok*I restriction endonuclease domain [25]. Each zinc-finger specifically interacts with three base pairs (bp) of the genomic target sequence and several zinc-fingers can be consecutively assembled to recognize and bind a total of 9 to 12 bp of DNA [26]. Zinc-finger nucleases must always be used in pairs, since their *Fok*I endonuclease domain is only catalytically active if it is present as a dimer [25]. The target motifs on the DNA are selected in such a way that the two zinc-finger nuclease monomers bind to the target DNA in antiparallel orientation and at a suitable distance to each other. A DNA double-strand break (DSB) is then catalyzed in the interspace between the two binding sites [27,28]. In addition to the particularly complex production process of zinc-finger nucleases, there are limitations in the selection of possible binding sites, as well as unpredictable neighbor effects between adjacent zinc-fingers on the DNA binding specificity [22].

In 1989, it was shown for the first time how transcription activator-like effector (TALE) proteins are transferred from a pathogenic bacterium of the genus *Xanthomonas* into a host plant [29]. The TALE-based nucleases (TALENs) act, similarly to zinc-finger nucleases, as dimers [30]. The DNA-binding domains consist of up to 30 copies of highly conserved repeats of typically 34 amino acids. Only the amino acid positions 12 and 13 (called repeat-variable di-residues) of each repeat are not uniform, because they specify the different DNA base pairs to bind [31]. A *Fok*I endonuclease bound to such a TALE binding domain induces the double-strand break in the target motif [30]. Since each repeat recognizes just one nucleobase of the DNA target, the design and assembly of the binding domain is more straightforward and versatile than with zinc-finger nucleases [31].

The platform of RNA-guided Cas endonucleases is derived from the CRISPR/Cas adaptive immune system of microbes. The most commonly used Cas endonuclease is from *Streptococcus pyogenes* (Sp). The two-component system consists of the Cas restriction enzyme and an artificial guide RNA (gRNA) that navigates the Cas protein to a cognate DNA sequence motif [12]. The gRNA consists of a structurally functional and a variable part. The 3′-tail forms a spatial structure required for binding with Cas to form a ribonucleoprotein complex. The variable part of the gRNA located at the 5′-end usually comprises 20 nucleotides and defines the DNA-binding specificity of the gRNA according to the principle of complementary base pairing. The genomic target motif of such ribonucleoprotein complex includes the same nucleotide sequence as the variable part of the gRNA, which enables the gRNA to specifically bind the opposite DNA strand. This part of the target DNA is often referred to as the protospacer, a term adopted from the microbial immune system these molecules originate from. In addition, this major part of the target motif is complemented by the so-called protospacer-adjacent motif (PAM), which is recognized and bound by the Cas protein. In the case of SpCas9, the nucleobase triplet NGG, in which N stands for any of the four nucleobases, represents the PAM site [12]. The high versatility of the gRNA 5′-end allows a wide variety of target sequences of choice to be addressed. The comparatively simple application and the efficiency and reliability achievable in higher organisms have rendered this platform the most popular and frequently used tool for site-directed genome modification today [32,33]. 

## 2. Methodological Aspects of Cas Endonuclease Technology

### 2.1. System Components 

When employing Cas endonuclease technology in plants, there are a number of opportunities and some specific requirements in terms of the construction of transformation vectors. Cas9 endonucleases have been modified in various ways for use in plants. Importantly, the coding sequences were complemented by one or two nuclear localization signals (NLSs) and the codon usage of the protein biosynthesis was optimized for use in plants in general, for monocots and dicots, or even more specifically for individual species, such as wheat and barley [34,35,36]. 

Depending on the host organism, also various promoters have been used to drive endonuclease expression. In crop plants, the doubled enhanced cauliflower mosaic virus (2x35S) promoter has been mainly used for test systems [34,37,38] and UBIQUITIN promoters for the generation of heritable modifications. Accordingly, the maize POLYUBIQUITIN1 promoter (ZmUBI1) has been commonly used for cas9 expression to generate heritable changes in barley and wheat [37,39,40]. Alternatively, Cas endonuclease-encoding genes can also be expressed by self-replicating virus particles in the plant cell [41]. This expression system ensures a comparatively high gene dosage, which may be particularly useful for homology-based approaches that are described in more detail below. Gil-Humanes et al. [42] have shown that the Wheat Dwarf Virus, a replicon-based geminivirus, can be modified to express gRNA and cas9 in wheat, albeit this approach has not yet been shown to be applicable for the generation of plants with heritable modifications.

The expression cassette for a gRNA typically consists of plant-derived RNA polymerase III (Pol III)-processed promoters and terminators derived from small nuclear RNA (snRNA)-encoding genes. Somehow surprisingly, it also proved to be sufficient in plants to use the transcriptional termination signal of bacterial CRISPR RNAs, which solely consists of a stretch of five to seven thymidines [34,43]. A comparative test in maize protoplasts had shown that wheat and rice U3 promoters were more efficient than the *Arabidopsis* U6 promoter that has been preferentially used in dicots [44]. The wheat U6 promoter has hitherto been mostly used for barley and wheat [37,39,40]. More recently, a study of Kumar et al. [45] indicated that the barley U3 promoter might be more efficient in site-directed mutagenesis of barley than the frequently used rice U3 promoter.

The preference of the U3 and U6 promoters for A and G as the first transcribed nucleotides limits the selection of target motifs to GN20GG and AN20GG, respectively [43]. However, there are thus far no examples where gRNAs were directly driven by Polymerase II-compatible promoters. The mRNAs otherwise expressed by Pol II promoters are processed at both ends and these modifications significantly alter the structure of the gRNA, and thus compromise the ability to bind both Cas9 and the target motif that is to be processed with sufficient efficiency [43].

Several systems have been developed for the expression of multiple gRNAs. Mostly, each gRNA is expressed by a separate Pol III promoter [44,46,47]. Alternatively, multiple gRNAs can be located on only one transcript and post-transcriptionally separated, as has already been demonstrated with wheat [48]. However, such expression systems have not yet led to improved efficiency of the technology with regard to individual target motifs in plant genomes [43,49,50]. 

### 2.2. Criteria for Target Motif Selection and in Silico gRNA Design 

Thanks to the opportunity to equip the gRNA with any target-specific 5′-sequence, the plant genomic target site of the gRNA/Cas complex can, in principle, be chosen at will. However, the Cas9 endonuclease requires the aforementioned two guanines that are part of the protospacer-adjacent motif (PAM) at the 3′-end of the target sequence. In addition, there are further preferences of the target-specific part of the gRNA, some of which can have a considerable impact on the functionality of the gRNA/Cas9 complex. For instance, it was determined for the so-called seed region, that is, the six nucleotides residing immediately upstream of the PAM site, that a GC content above 50 percent increases the probability of sufficient gRNA functionality [51]. With regards to the entire target motif, an enrichment of guanines and low adenine content was shown to cause increased binding stability and activity of the gRNA/Cas9 complex [52]. 

The target sequence-specific part of the gRNA usually has a length of 20 nucleotides [12,53]. As previously stated, the transcriptional start defined by the U3 and U6 promoters, which are preferred to drive gRNAs in plants, are A and G, respectively, which requires the selection of genomic targets having the same nucleobases at their 5′-end [38]. In human cell lines and zebrafish, it has been shown that the length of the target-specific part of the gRNA can be shortened to 18 nucleotides without a noticeable effect on mutation efficiency. Surprisingly, such shorter gRNAs even had increased specificity for the on-target sequence as compared with off-targets [52,54,55]. Guide-RNAs with less than 20 target-specific nucleobases also showed high performance in *Arabidopsis* and barley [56,57]. On the other hand, an extension of the gRNA 5′-part beyond 20 nucleotides was reported to cause reduced cleavage efficiency [58].

Quite a number of online platforms have been developed for the selection of target motifs and corresponding gRNAs. For instance, CRISPR-Plant was especially developed for plants [59,60]. While a whole array of plant reference genomes can be directly screened in this platform, the barley and wheat genomes are not implemented. However, for the selection of gRNAs in cereals, DeskGen is a highly instrumental alternative, as these cereal genomes are available for target validation [61,62]. This platform offers a comparatively large scope of options, including various Cas endonucleases and a useful range of lengths of the target-specific gRNA 5′-part. Also, the CRISPOR online tool can be specifically used for barley and wheat. It comprises a comprehensive choice of different Cas enzymes [63,64]. In addition to the selection of gRNAs predicted to be well-performing at their target motif, all above online tools are also capable of indicating potential off-targets in the chosen reference genome. WU-CRISPR is another useful platform, as its algorithm is comparatively strict in selecting particularly useful target motifs. However, this tool is confined to SpCas9 and does not offer off-target screens in reference genomes [65]. A general disadvantage of all these platforms is that their algorithms have been established using data from other organisms than plants, which is thought to be one of the reasons that the reliability of their results is still fairly limited. 

Liang et al. [66] have focused on the preservation of the gRNA secondary structure. They determined that three of the stem loops frequently occurring in the gRNA 3′-part are essential to appropriately bind to the Cas9 protein, and hence are also of decisive importance to the overall functionality of the gRNA/Cas complex. These authors also found out that 98 percent of well-performing gRNAs have no more than a total of 12 base pairs formed between their target-specific and 3′-parts and no more than 7 target-specific nucleobases consecutively involved in intra-molecular base pairs. In addition, no more than 6 base pairs should be formed within the target-specific part of the gRNA. In order to ensure these gRNA features, it is recommended to thoroughly investigate the secondary structure of candidate gRNAs. For the prediction of RNA secondary structures, online platforms such as mfold [67,68,69] or RNAfold [70,71] are available.

In cases where genetic modifications are intended to be performed in genotypes other than those with reference genomes, it is recommended to countercheck pre-selected target motifs for their presence and integrity, because even single nucleotide polymorphisms typically result in a dramatic drop in gRNA/Cas efficiency. 

### 2.3. Delivery of Cas Endonucleases and Associated Reagents into Plant Cells 

A number of methods are available for the transfer of DNA, RNA, and proteins into plant cells, which in principle can also be used for Cas endonucleases, customized gRNAs, DNA repair templates, and any further reagents that may be used to support site-directed genome modification. Nowadays, the most widely used approach is based on the genomic integration of expression units with gRNA and Cas-encoding DNA sequences. However, the production of stably transgenic plants is a particular challenge for cereals, because agrobacteria, which are mostly used for plant transformation, have a very limited compatibility with these non-host plants. Moreover, the formation of adventitious shoots from leaf or shoot explants, which is readily achieved in the context of DNA transfer methods applicable to most dicotyledonous species, has only been successful in exceptional, hardly reproducible cases in cereals [72]. On the basis of special methodological approaches, routine genetic transformation of barley and wheat is nevertheless possible, at least by using selected accessions that are comparatively amenable to methods of DNA transfer. Such experimental model genotypes are, for example, the barley cultivar Golden Promise and the Mexican wheat breeding line Bobwhite SH9826 [11]. For stable DNA transfer in cereal crops, immature zygotic embryos are most widely used. Importantly, such embryos must be of a developmental stage at which highly totipotent cells are still abundantly present. In immature embryos of barley and wheat, such cells are preferentially residing in the area that surrounds the shoot apical meristem and the scutellum, respectively. Stable transgenic wheat and barley plants were already produced by ballistic transfer of plasmid-coated gold particles to immature embryos in the early 1990s [73,74]. Using hypervirulent bacterial strains, which had initially been used for the transformation of rice [75], thereafter it was soon also possible to generate transgenic barley and wheat plants using this principle [76,77]. Today, it is routinely possible to generate stable transgenic plants from over 10% of inoculated immature embryos of barley [78], while in wheat, efficiencies of about 5 percent have been achieved both via *Agrobacterium*-mediated and ballistic DNA transfer [79,80]. However, a largely improved protocol was more recently demonstrated to allow for much improved transformation efficiency in wheat ([81,82]. The comparative examination of a variety of genotypes has shown that although some can indeed be transformed in addition to the model lines, the efficiency, however, is much reduced [83,84]. Previous studies on Cas9-induced mutagenesis of barley have been based without exception on *Agrobacterium*-mediated DNA transfer. In contrast to this, site-directed mutagenesis in wheat has mostly relied on ballistic gene transfer [85]. 

Another particularly interesting principle of DNA transfer into cereal cells is the use of immature, single-celled pollen (microspores), which can undergo cell proliferation and embryogenic development under suitable culture conditions [86]. Since pollen consists of haploid cells, it is possible that homozygous transgenic plants can be directly produced via this developmental pathway in association with DNA transfer and whole genome duplication. *Agrobacterium*-mediated DNA transfer in embryogenic pollen cultures of barley has the further advantage that a winter barley variety can be used, which is comparatively difficult via DNA transfer to immature embryos [87,88]. A similar transformation method has been reported also for wheat, which is, however, based on ballistic DNA transfer [89]. In the authors’ laboratory, barley mutants have already been produced via the pollen embryogenesis pathway using TALENs [90], as well as by Cas endonucleases. Furthermore, the production of stable transgenic barley and wheat by ballistic DNA transfer into meristematic tissue from the shoot apex of embryos prepared from mature grains was also successful [91,92]. This method has potential because of particularly low genotype dependence, since plant regeneration is based on conventional germination of the embryos, and therefore no formation of adventitious shoots is necessary. More recently, Hamada et al. [93] have also demonstrated that the production of plants with site-directed modifications using gRNA/cas9 constructs is possible using this method.

After alteration of the genomic target motif, the presence of transgenes coding for gRNA and endonuclease is not only unnecessary but also undesirable, as off-targets can still be mutated, even if they are not identical with the on-target. In this context, an advantage of the application of Cas endonucleases compared to conventional genetic engineering is that the integration site of the transgenes is mostly not coupled with the site of the desired genetic modification. Thus, the elimination of these transgenes by genetic segregation of progeny is comparatively straightforward while maintaining the desired modification. Endonuclease-triggered genetic alterations in primary transgenic plants are often heterozygous and restricted to tissue sectors. Therefore, progeny needs to be screened in order to select homozygously mutated individuals. Consequently, no extra time is required to obtain transgene-free mutants. Haploid technology is a particularly elegant solution for the separation of achieved genetic modifications from unnecessary transgene insertions or for the segregation of several mutations present in chimeric plants of the M1 generation. With this technology, populations of completely homozygous recombinants can be obtained, in which desired genotypes are present at a much higher frequency than among offspring derived from conventional selfing [94,95].

In addition to the various possibilities of genomic integration of gRNA and cas endonuclease genes, methods for transient expression have also been established, which is of considerable value for the preliminary validation of target-specific gRNAs, Cas endonuclease variants, and any further components used in this context. The most widely used transient expression method is based on the transfection of isolated mesophyll protoplasts, whose plasma membrane is rendered porous by application of polyethylene glycol. This enables transgenic plasmid DNA, in vitro transcribed RNA, bacterially pre-produced Cas protein, or ribonucleoprotein complexes to be taken up by the protoplasts. This has been exemplified, amongst others, in wheat and barley [39,57,96]. The activity of the transferred components can then be checked after amplification of the genomic target regions using the T7E1 assay, by Sanger or deep sequencing, as is described in more detail further down below. Another method for the functional validation of gRNA/cas constructs by transient expression is based on the ballistic transformation of leaf epidermis. Epidermal cells are comparatively well suited to be screened microscopically, for instance in regards to Cas-induced restoration of a reporter gene construct [97]. Barley leaf tissue has been used to demonstrate precise editing via homology-based DNA repair using a synthetic DNA repair template [98]. 

As an alternative to the expression of gRNA- and Cas-encoding DNA, in vitro transcribed RNA or Cas protein can be transferred to plant cells in order to make specific modifications to the genome. Guide RNA and Cas endonuclease can also be pre-assembled to form gRNA/Cas9 ribonucleoprotein complexes prior to their transfer into plant cells. The use of pre-produced RNA and protein molecules or complexes thereof has the advantage over DNA that their effective amount is independent of the expression profile and strength of plant promoters. In addition, the activity of these components is limited in time, as they are not continuously delivered by gene expression but are subject to cellular degradation. Accordingly, fewer mutations will occur in off-target motifs during the subsequent development of the plants. A further advantage is that mutated progeny do not have to be examined for the loss of integrated gRNA- and Cas9-encoding DNA sequences, as all mutated individuals can be used without restriction due to their transgene-free nature. In wheat, the GW2 gene was used as an example to show that the transfer of in vitro transcribed RNA coding for both gRNA and Cas9 leads to mutations in one percent of ballistically transformed cells [99]. About one third of the resulting plants were mutated in all six copies of the hexaploid genome. Compared to the transfer of gRNA and cas9 transgenes, however, this method had a mutation rate that was about 60 percent lower. Liang et al. [96] achieved an improvement in mutation efficiency by assembling gRNA and Cas9 into ribonucleoprotein complexes before ballistic transfer into immature embryos. After selection-free plant regeneration, mutants were identified with a frequency of more than four percent, which is on a par with the efficiency of conventional wheat transformation. 

### 2.4. From Site-Directed Mutagenesis to Precise Genome Editing

Current utilization of Cas endonuclease technology is still largely limited to random alterations of the DNA sequence at the user-defined genomic sites. This comparatively simple approach of site-directed mutagenesis relies on the error rate of the DNA repair mechanism, called non-homologous end-joining (NHEJ), which is predominant in plant cells. In this process, the two DNA ends resulting from a double-strand break are recognized and relegated irrespective of their nucleobase sequence. 

Increased predictability of site-directed modifications was shown to result, for example, from two simultaneously induced DNA breaks, which can lead to the precise deletion of the interjacent fragment [22]. In addition, microhomology-based DNA repair produces predictable deletions in a comparatively simple way, provided that identical sequence repeats are present at both DNA ends that are to be relegated [100].

A more ambitious approach is the so-called base editing, in which a single nucleotide is specifically converted into another so that no more than one amino acid of the encoded protein is altered at a time [101]. For this purpose, a Cas-based nickase, that is, a Cas endonuclease derivative that cuts only one strand of DNA due to the mutative alteration of one of its two nucleolytic domains, is bound to a natural or artificial nucleobase deaminase enzyme [102]. Cytidine deaminases can convert C/G basepairs to T/A in the target region, whereas adenosine deaminases induce A/T to G/C conversions [102,103]. The functionality of cytidine and adenosine deaminases has already been demonstrated in several plant species, including wheat [104,105]. The effective base editing area ranges in dependence of the base editor used, e.g., the cytidine deaminase used by Zong et al. [104] can convert positions 1 to 17 within the target site and the adenosine deaminase used by Li et al. [105] is capable of converting positions 4 to 8.

In addition to the aforementioned error-prone non-homologous end-joining (NHEJ) DNA repair mechanism that predominates in plant cells, homology-directed repair (HDR) can also be used to generate precise genome alterations, albeit this process is much less active in somatic cells. Homology-directed repair naturally involves the sister chromatid of the same chromosome or the homologous chromosome as the correct sequence template, allowing the original sequence to be restored, even in cases of comparatively severe DNA damage. By using artificial DNA repair templates that are partially complementary to the site of the plant genome to be modified, it is possible to make even fairly large modifications precisely, as specified by the experimenter [106]. However, due to the methodological challenges, there are very few examples of this precise editing in plants published thus far. A first experimental approach involved the stable integration of the repair template together with the gRNA- and Cas-encoding expression units into the plant genome. The integrated repair template is flanked by the target sequences of the gRNA-mediated endonuclease used, so that it is cut out by the Cas restriction enzyme. This principle has proven to be sufficient in *Arabidopsis* [107]. The use of paired Cas9 nickases, which generate two single-strand breaks on the two opposite DNA strands, has increased the efficiency of homology-directed DNA repair [50,108,109]. With geminivirus replicons as carriers of the repair template, the dose per cell is increased. In this case, a strain of the Bean Yellow Dwarf Virus was modified so that only the elements essential for the method remained and the artificial repair template was amplified with high frequency [110]. In barley and wheat, precise editing has been achieved either only at the cellular level [98] or has the limiting prerequisite that the obtained genetic modification leads to an in vitro selectable trait, e.g., an herbicide resistance [111]. 

### 2.5. Identification and Characterization of Site-Specifically Modified Plants

The presence of gRNA- and Cas-encoding expression units is usually tested by PCR amplification [112,113]. In some studies, the expression of gRNA and cas9 was additionally assessed by RT-PCR, which can be very informative, especially in the process of method establishment [37]. For the detection of site-directed modifications, PCR products derived from the genomic target region can be analyzed using various methods. Digestion of the PCR product using conventional restriction enzymes is the most straightforward approach. However, the prerequisite for this is that the restriction site of the Cas endonuclease lies within a recognition sequence of the conventional restriction enzyme that then can be used for the test. In contrast to the wild-type sequence, PCR products of mutated target motifs cannot be digested by this enzyme [34]. A disadvantage of this method is that the choice of target motifs is substantially limited. This can be circumvented using the T7E1 assay, in which a mixture of PCR products of mutant and wild-type alleles is digested by the mismatch-sensitive T7 endonuclease I [39,114]. Another disadvantage these methods have in common is that their sensitivity is low, i.e., if mutated alleles of a plant are limited to small tissue sectors, detection may be not possible. This leads to a certain number of mutants not being identified. 

To avoid this problem, Liang et al. [115] used target-specific synthetic gRNA and Cas9 protein for in vitro digestion of PCR products from genomic targets. This method is independent of conventional restriction sites in the target motif and was reported to be extremely sensitive. However, a large-scale application of this approach is hampered by the fact that recombinant Cas protein is fairly expensive.

As compared with the aforementioned restriction-based assays, sequencing of PCR product derived from the genomic target region is more informative. In heterozygous or chimeric mutated plants, the sequencing chromatogram shows double and multiple peaks at the individual base positions, which usually starts at the restriction site of the Cas endonuclease and continues in the same direction as the sequencing was conducted. Various bioinformatic tools, such as Tracking of Indels by DEcomposition (TIDE) [116] and DSDecode [117], provide the opportunity to read out such peaks, thereby determining the different allele variants present in the analyzed plant sample. A more reliable approach involves the analysis of several single clones via Sanger sequencing, as this provides a detailed breakdown of the mutations present in the analyzed sample. In addition, more conclusive information on the proportions of the mutated alleles can be obtained. In this context, it is important to consider that Cas endonuclease-triggered mutations are usually heterozygous and often limited to sectors of the plant. Moreover, leaf samples are not necessarily representative for the whole plant. This phenomenon is reflected by the observation that not all mutations detected in primary mutants are inherited by subsequent generations. In addition to Sanger sequencing, deep sequencing methods can be used to analyze mutations in individual plants, which is particularly informative owing to the obtained amount of data [48,118]. In practice, it is typically necessary to find a reasonable compromise between workload, conclusiveness of assays, and costs. Therefore, a combination of different methods is often the most suitable solution [40,42]. 

The molecular analysis of candidate plants can also address accidental modification of so-called off-target sites, which can cause undesired side effects and also influence the efficiency of on-target mutagenesis. In plants, the proportion of off-target mutations is typically below 1% [119]. In contrast to human cells, this does not pose a serious problem. These mutations are often confined to small sectors that are usually not passed on to offspring. Furthermore, due to the high specificity of the gRNA/Cas complex, possible off-target sites are well predictable so that progeny can be readily tested to eliminate segregants with unwanted mutations. 

Since primary mutants are often chimeric or heterozygous and integrated cas9 and gRNA transgenes can be inherited, the analysis of progeny is essential to identify homozygous mutants that are null segregants in regards to the transgenes [35,112]. 

It is not recommended to spend too much effort for phenotypic analyses of primary mutants, because in addition to their uncertain genetic homogeneity and zygosity, the general fitness of these individuals is typically strongly affected owing to the manipulation and culture procedure they have gone through [120]. Importantly, it is advisable to use non-transgenic, non-mutant segregants as wild-type controls for the analysis of plant traits, because the comparison with other plants bears a fairly high risk of obtaining misleading results. A survey of the general workflow of targeted genome modification in barley and wheat is depicted in Figure 1. 

## 3. Applications

While in this section only some representative examples for applications of Cas endonuclease technology in barley and wheat are presented in more detail, Table A1 provides a comprehensive overview of studies hitherto published on these crops. 

In 2013, Upadhyay et al. [37] were the first to show that Cas endonucleases are in principle applicable in a *Triticeae* species and especially in the large and complex genome of wheat. However, this study was still confined to site-directed mutagenesis in a cell suspension, from which no plants can be regenerated. Mutations were detected in 18 to 22 percent of the sequenced amplicons derived from target regions of the INOSITOL OXYGENASE (TaINOX), and PHYTOENE DESATURASE (TaPDS) genes. The first plants carrying Cas endonuclease-induced mutations were presented by Wang et al. [39]. While heritable loss-of-function mutations were induced in all three homoeologues of the MILDEW-RESISTANCE LOCUS O (TaMLO) gene by the use of TALENs, Cas9-triggered mutations were still confined to the A subgenome. Using two wheat backgrounds, the authors produced a total of 687 gRNA/cas9 primary transgenic plants, of which about 5 percent proved to carry mutations in the target motif. However, in contrast to the TALEN-induced ones, the heritability of these mutations was not shown. 

The first published use of Cas endonucleases in barley aimed to induce mutations in HvPM19 [35], which encodes an ABA-induced plasma membrane protein previously described in wheat as a positive regulator of dormancy. Four copies of this gene are present in barley. For two of those, knockout mutants were generated. Amongst 13 primary transgenic plants, three carried mutations in HvPM19-1, whereas one HvPM19-3 mutant was found amongst ten T0 plants. While this study was the first to provide evidence for the heritability of Cas9-induced mutations in a *Triticeae* crop, a resultant phenotype was not described.

Once the applicability of Cas endonuclease technology to achieve heritable mutations was shown for barley and wheat, this principle was used for many further approaches. Holme et al. [40] used Cas endonuclease-induced mutations to investigate the function of the PHYTASE GENE A of barley. HvPAPhy_a acts as the main regulator of phytase content in the barley grain. Phytases are important sources of bioavailable phosphorus, and therefore particularly important for germination. To reduce the activity of HvPAPhy_a independently of HvPAPhy_b, which is very similar in its gene sequence, Holme et al. specifically mutated the promoter of HvPAPhy_a. While small insertions and deletions in homozygous mutant progeny had only little effect, larger deletions, which also affected exon 1 of the target gene, caused a significant reduction of phytase activity. 

Taking a Cas9-mediated knockout approach, Wang et al. [118] investigated the function of the GW2 gene of wheat. The rice orthologue of TaGW2 had been previously described as a negative regulator of grain size and thousand-grain weight. Wang et al. produced knockout mutants for all three subgenomes of wheat by RNA-guided Cas9 and generated all possible combinations of mutated homeoalleles by crossing. The knockout of individual TaGW2 homeologues in any of the subgemomes increased the size of the wheat grain, as well as the thousand-grain weight. Consequently, the authors concluded an additive effect of all three TaGW2 homoeologues [118].

Gluten in the wheat grain is relevant for the baking properties of the flour. This protein fraction is encoded by about 100 genes. Gluten intolerance is mainly based on the immune response to a particular peptide within the α-gliadins, which are encoded by 45 genes. Sánchez-León et al. [113] designed two guide-RNAs specific to the gene region encoding the hyperallergenic epitopes and produced bread and durum wheat lines with knockout mutations in up to 33 of the 45 α-gliadin genes. Thus, immunoreactivity was reduced by as much as 85 percent. The inheritance of the mutations and their effect on the α-gliadin content were followed over three generations. In order to further reduce the immunogenic α-gliadins in wheat, further work is still necessary [113].

In barley and wheat, Cas endonucleases have been mainly used for the investigation of gene functions by knockout, for the modification of metabolite contents and to increase resistance to fungal pathogens (Table A1). These studies show the great potential of this technology for research and breeding. In addition to the generation of random sequences at the target site, more recent work has demonstrated predictable nucleobase exchanges in wheat by the use of a chimeric Cas9 derivative, which consists of a nickase and an adenosine deaminase [105]. This base editing approach opens up further possibilities of generating functional gene variants rather than knockout alleles.

## 4. Regulation

Parallel to the development of the Cas endonuclease technology, science organizations, politicians, and society are dealing with the effects of the technology. The following section will briefly address some related aspects, in particular the regulation of plants that carry site-directed mutations. The focus here is on the USA and Europe, as these two have very contrasting approaches.

For the United States, Wolt and Wolf [121] summarized the regulation of the so-called green biotechnology, and in particular the use of customizable endonucleases. Commencing at the Asilomar conference on biosafety in 1975, a regulatory framework for plants with recombinant DNA has been developed in the USA since the mid-1980s. Over the years, this framework became more and more complex due to more accompanying research and the increasingly lengthy periods needed for decision-making. Although the focus of regulation in the United States is on the product, the procedure also includes process-based facets. For example, genetically engineered plants have to undergo special approval procedures. However, these differ between plants that have been transformed by use of the plant pathogen *Agrobacterium* and those that have been ballistically transformed. If plants with an identical trait are produced without genetic engineering, no special regulation takes place. For plants bred by use of customized endonuclease technology, the portal “Am I regulated?” has been established, in which the USDA made case-by-case statements at short notice as to whether a plant needs to be regulated or not. In 2017, work began on revising the regulations on biotechnology. The major trigger for regulation was shifted towards the process by a redefinition of “genetic engineering” as “mean techniques that use recombinant or synthetic nucleic acids with the intent to create or alter a genome”. This is not applied to processes of targeted genome modification that cause deletions or base edits or result in the targeted insertion of a DNA fragment, which would also be possible by conventional breeding, conventionally induced mutagenesis or even without any human intervention [121].

In a detailed report in 2017 [122], the European Academies’ Science Advisory Council (EASAC) stressed the possibilities that targeted genome modification offers. In plant breeding, improved precision compared to undirected mutagenesis methods represents a significant advance. At the same time, some representatives of non-governmental organizations and political parties are viewing biotechnological breeding methods critically. According to EASAC, this gives science a special responsibility to explain its work. In regards to plant breeding, EASAC recommends to politicians not to regulate transgene-free plants carrying site-directed modifications as being “genetically engineered” in the sense of the EU Directive 2001/18/EC on the deliberate release into the environment of genetically modified organisms. Additionally, it advises a revision of the European genetic engineering legislation with a stronger focus on the product instead of the breeding process. The need for international compatibility of regulation is emphasized and reference is also made to best practices from other countries, such as the United States, where site-directed mutagenesis is not regulated as genetic engineering [122]. 

While some countries, including the United States, Argentina, Brazil, Chile, and more recently Australia, have already made decisions to regulate targeted mutagenesis on a case-by-case or general basis not differently than conventional methods of plant breeding, in Europe, those varieties are the subject of the Directive 2001/18/EC. According to that, field trials with plants carrying targeted mutations need special permission on application, and placing respective varieties on the market needs tedious and very costly approval procedures. The European plant research community has responded to this with demands for a revision of European genetic engineering legislation. For more details see Sprink et al. [123] or the position paper of 95 European plant research institutions (http://www.vib.be/en/news/Documents/Position%20paper%20on%20the%20ECJ%20ruling%20on%20CRISPR%2012%20Nov%202018.pdf).

In many countries in the world, decisions on how to regulate plants generated using customized endonucleases are currently pending. They will play a key role in deciding how and by whom these technologies and their possibilities can be used.

## 5. Perspective 

The new possibility of utilizing customized endonucleases in barley and wheat opens up a broad spectrum of opportunities for applications. A generally limiting aspect, however, is that the genetic transformation of cereals poses a particular challenge. In the context of the question for new or improved methods of transferring DNA into regenerable cells of cereals, it has to be considered that a number of methods have already been developed, but have not yet been utilized for the application of Cas endonucleases. A remarkable example of this is the use of isolated microspores or embryogenic pollen cultures generated from them for DNA transfer using *Agrobacterium* [87], ballistics [89] or electroporation [124]. When it comes to the development of new methods, particular attention must also be paid to the fact that they should be applicable to a broad spectrum of genetic backgrounds, which unfortunately does not apply to any of the methods used to date. On the other hand, the particular advantage of Cas endonuclease technology will only be fully exploited if it is possible to directly modify any plant genotype of choice. This is the only way to completely overcome the cumbersome introgression of advantageous gene variants into elite germplasms, and above all, the genetic coupling of desired alleles with undesired ones. In this context, the dissection and culture of ovules is another remarkable possibility for the in vitro regeneration of wheat and barley [125,126]. For the *Agrobacterium*-mediated DNA transfer into cultivated ovules, in which the T-DNA is transferred into few-celled proembryos, it was shown that this method has comparatively low genotype dependence [127]. 

In all studies published to date in which heritable site-directed modifications have been achieved in the barley and wheat genomes using Cas endonuclease technology, the POLYUBIQUITIN1 promoter from maize was used to express the cas9 gene. For the stable expression of protein-encoding transgenes in the *Triticeae*, however, a vast number of further promoters with different activity profiles are available, some of which may also be used for Cas endonucleases [128,129,130]. In addition, there is, thus far, unexploited potential for the expression of gRNAs in barley and wheat. For instance in rice, a t-RNA processing-based system was established, in which multiple gRNAs are released from a complex transcript by endogenous RNases [50,131]. 

To keep pace with the frequent emergence of novel Cas and gRNA formats, as well as regulatory and further functional elements, several modular vector systems have been developed that allow for a rapid and versatile assembly of any combination of components of choice. Some of these systems may also be useful for the application in the *Triticeae* cereals [45,50].

Whereas site-directed mutagenesis represents the currently well-established state of the art of Cas endonuclease technology, more precise techniques of targeted genome modification have unfortunately not yet become routine, and are at best exemplary in plants. A portfolio of chimeric Cas derivatives is now available for the so-called base editing using nucleoside deaminases, which allows conversions from G to A, C to T, T to C, and A to G. Thus, the vast majority of spontaneous point mutations occurring in nature can be specifically generated, or if desired, the corresponding wild-type alleles can be restored as well [103]. A broad application of precise genome editing in association with DNA repair templates, on the other hand, still poses particularly great methodological challenges. Homology-based repair, for example, is comparatively rarely occurring, which entails a correspondingly low efficiency of conventional selection marker genes whose genomic integration and expression is too poorly associated with template-mediated repair events. In addition, the artificial repair templates must compete with the naturally recruited homologous sequences present in the cell on sister chromatid and homologous chromosome(s), which further reduces the efficiency of targeted modification. 

A further limitation of Cas endonuclease technology is that the binding site of the gRNA/Cas complex cannot be predefined entirely at will. The most commonly used enzyme for target sequence-specific genome modification is Cas9 from *Streptococcus pyogenes*. This enzyme requires the aforementioned protospacer-adjacent motif (PAM) NGG for binding to the genomic target. However, there are several other Cas9 variants that require other cognate DNA motifs. Such Cas9-orthologous enzymes are known from various representatives of the genera *Streptococcus*, *Staphylococcus*, *Campylobacter*, and *Neisseria* [132,133,134]. In *Arabidopsis*, for example, the *Staphylococcus aureus* and *Streptococcus thermophilus* Cas9 variants have already been used in addition to the standard SpCas9 [135]. Furthermore, in order to extend the flexibility in the choice of targets beyond the naturally occurring PAMs, Hu et al. [136] developed an artificial Cas9 variant. The advantage of this “expanded” Cas9 (xCas9) is that almost any nucleobase triplet is accepted for its binding to the genomic target. The synthetic Cas9-NG, which was developed in a different way and published shortly thereafter, proved to be even more efficient than xCas9 in many PAM variants [137]. 

While Cas9 endonucleases are widely employed in plants, Zetsche et al. [138] described another Cas endonuclease, namely Cas12a (Cpf1) from *Francisella novicida*, that can also be used to induce targeted alterations in genomes. Later, Cas12a-orthologous enzymes from other bacteria were also successfully used [139]. Besides some other peculiarities, Cas12a, in contrast to Cas9, produces 5′- overhangs rather than blunt DNA ends. This provides a particularly attractive option for the establishment of robust methods of precise genome editing using homology-directed DNA repair. 

Cas endonucleases have also been modified in such a way that they induce single-strand breaks in double-stranded DNA. Among such nickases, there are specific variants for each of the two DNA strands [12,140]. In *Arabidopsis*, paired nickases derived from Cas9 were used to demonstrate that both the specificity of the binding to the genomic DNA and the efficiency of homology-directed DNA repair can be increased [50,108,109]. 

Whereas knockouts are the standard outcome of site-directed mutagenesis, the possibility to achieve quantitative changes in the functionality of target genes has not yet been exploited in cereals. Using tobacco as experimental model, Schedel et al. [141] have shown that in-frame mutations can maintain gene functionality to a reduced extent. This can be of particular value for the targeted modification of genes that are essential for the plant or the generation of new allelic diversity for the use in plant breeding. In a particularly elegant approach, Rodriguez-Leal et al. [142] produced tomato lines with various CLAVATA3 alleles by simultaneous expression of several gRNAs specified for different target motifs in the promoter of this gene. The resultant gene variants led to a whole series of lines with different fruit sizes according to the individual strength of CLV3 expression. 

In other crop plants, there are various application examples of customizable endonucleases, which may also be considered for breeding programs in the *Triticeae* cereals. Since every characteristic of plants is largely genetically determined, it must in fact be possible to improve every trait that is relevant for the utility value of crops by employing Cas endonuclease technology. Due to these extensive possibilities, only a few of the promising options can be mentioned here as examples. A potential field of application for Cas endonucleases is the establishment of resistance to various potyviruses, which has already been achieved, for example, in cucumber. The established resistance relies on the knockout of the host plant’s EIF4E gene, which is essential for the translation of viral mRNAs [143]. Resistance-conferring alleles are also known from a related barley gene [144,145]. In addition, there are a number of susceptibility genes for diseases caused by other pathogens, which is similarly suitable for resistance breeding. 

The ability of Cas endonucleases to completely eliminate the function of a gene seems particularly attractive when it comes to freeing crops from toxic, carcinogenic, allergenic, or bad-tasting gene products or metabolites. The aforementioned study by Sanchez-Léon et al. [113] on the production of wheat with gluten, whose fraction of immunodominant gliadins is drastically reduced, is a promising example in this respect. Another potential use of Cas endonuclease technology results from the fact that most modern durum wheat varieties carry a mutation in a heavy metal transporter gene, which entails a reduced sequestration of cadmium in the roots so that this toxic element accumulates in the grain to an extent that is of concern in terms of human health [146]. The restoration of the corresponding wild-type allele would, therefore, be a particularly promising way of improving the quality of pasta, which is widely consumed across the entire globe. 

As has already been demonstrated in potato [147] and rice [148], it is also possible to modify the starch quality by specifically modulating the biosynthesis of the major starch components amylose and amylopectin. While amylose-free starch is of particularly high value for the paper and chemical industries, starch with reduced amyplopectin content is referred to as resistant starch, which, due to its fiber-like properties, has the potential to counteract type-2 diabetes, a civilization disease that is now increasingly spreading even in developing countries. 

For future approaches, it is also conceivable to modify many different genes in one step, for instance to influence complex processes such as photosynthesis [149]. Cas endonuclease technology offers unprecedented prerequisites for this due to the option of simultaneous use of numerous gRNAs. 

In summary, it can be stated that the establishment of Cas endonuclease technology represents a quantum leap for plant research and breeding. Even with regard to major methodological challenges, the current pace of research and development in this field means that further very useful solutions are highly probable in the near future. The implementation of this technology, especially in the agronomically important cereals of the *Triticeae* cereals, will likely continue to gain momentum, and is thus expected to increasingly contribute to the effective production of sufficient and high-quality food, feed, and industrial raw materials, while ensuring largely improved environmental compatibility of agriculture. 

## Figures and Tables

**Figure 1 ijms-20-02647-f001:**
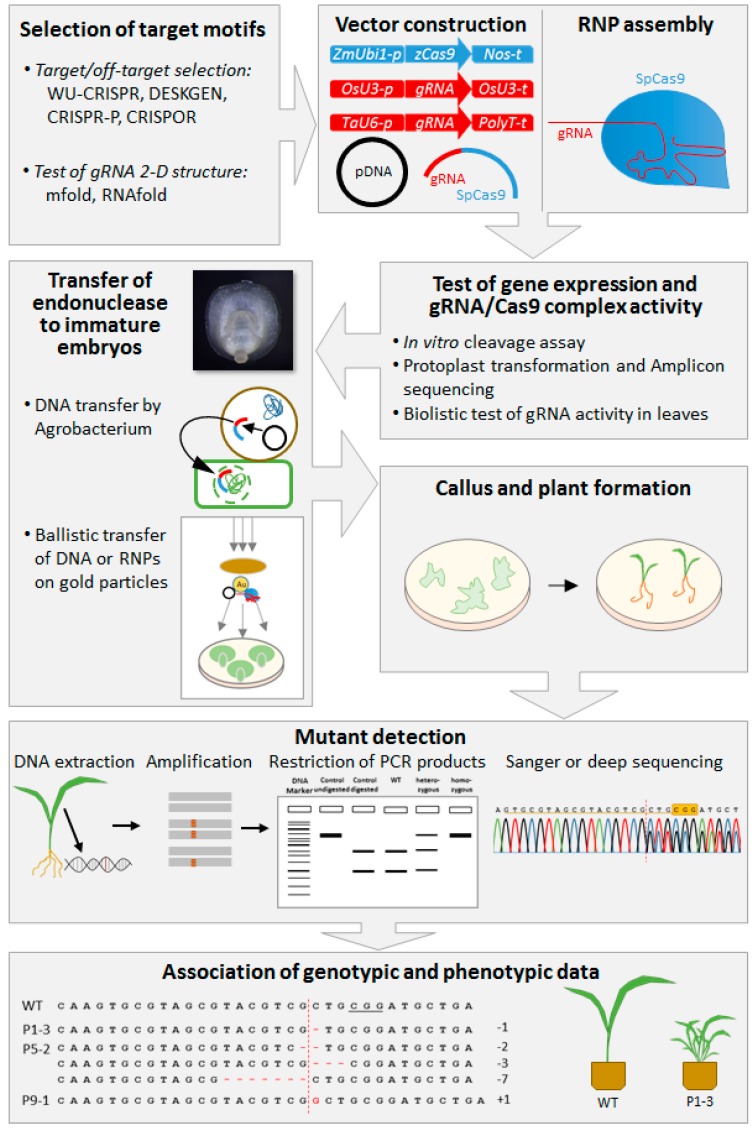
General workflow of targeted genome modification in barley and wheat, including target motif selection, vector construction, vector tests, gRNA and Cas9 delivery, plant regeneration, mutant detection, and genotypic and phenotypic analyses.

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
