# Peer review of "Cas Endonuclease Technology—A Quantum Leap in the Advancement of Barley and Wheat Genetic Engineering"

_ijms, 2019, doi:10.3390/ijms20112647_

Round 1

Reviewer 1 Report

The authors have done great job in providing historical and recent applications of Cas endonuclease technology and provided their detailed perspective on it futher possibility for wheat and barley improvement. I would recommend this mansucritp to be accepted after correcting some grammatical errors/typos such italicized scientific names, and correctly format the referencing system based on authors guidlines. 

Author Response

scientific names have been changed to not italicised format and the referencing system was reformated according to the journal's guidlines

Reviewer 2 Report

This is a well written manuscript which comprehensively outlines the state of art in this complex field in a lucid way.

Just a few minor editorial remarks:

l. 32-43 This first section does not really meet the header of 1.1. Rather, It presents general statements which apply to merely any agricultural approach (and which, simplistic as they are, may appear a bit tiring). Furthermore, there meanwhile is a controversial discussion among experts whether the world's population will grow or decline in the foreseeable future.

l. 214 Better skip "unfortunately" at this place.

l. 390 "conducted": demonstrated, checked?

Chapter 5. Just a semi-serious comment: Some of these potential applications (starch, allergens) I heard about some 30 yrs. ago when the hymns were sung to the potentials of "classical" genetic engineering ;-)

That's it.

Author Response

Reviewer: l. 32-43 This first section does not really meet the header of 1.1. Rather, It presents general statements which apply to merely any agricultural approach (and which, simplistic as they are, may appear a bit tiring). Furthermore, there meanwhile is a controversial discussion among experts whether the world's population will grow or decline in the foreseeable future.

Authors' response: Some 'tiring' redundancy was removed from the text and the current growth of the world's population was expressed in a more general way to render the text less speculative.

Reviewer: l. 214 Better skip "unfortunately" at this place.

Authors' response: done

Reviewer: l. 390 "conducted": demonstrated, checked?

Authors' response: 'conducted' was exchanged by 'checked'

Reviewer: Chapter 5. Just a semi-serious comment: Some of these potential applications (starch, allergens) I heard about some 30 yrs. ago when the hymns were sung to the potentials of "classical" genetic engineering ;-)

Authors' response: We entirely agree that we have to be very careful when making statements that may be interpreted as promises, which has all too often let to difficulties in the public discurse on plant genetic engineering. Nonetheless, it needs to be allowed to express visions and, most importantly, Cas endonuclease technology has an unprecedented potential to facilitate such solutions, because, in contrast to RNAi technology, it is capable of producing true and durable gene knockouts.